# Gradient Manifold Geometry as a Signature for Adversarial Detection

## Abstract

Despite their remarkable performance, deep neural networks exhibit a critical vulnerability where small adversarial perturbations can drastically alter predictions, making robust detection paramount for safety-critical applications like autonomous driving. To address this, this paper investigates the geometric properties of a model's input loss landscape by analyzing the Intrinsic Dimensionality (ID) of the gradient parameters, which quantifies the minimal number of coordinates required to describe data on its underlying manifold. We reveal a distinct and consistent difference in the ID for natural and adversarial data, which forms the basis of our proposed detection method. Our approach is validated across two distinct operational scenarios: in a batch-wise context for identifying malicious data groups on datasets like MNIST and SVHN, and more critically, in the individual-sample setting, where we establish new state-of-the-art results on challenging benchmarks such as CIFAR-10 and MS COCO. Our detector significantly surpasses existing methods against a wide array of attacks, including CW and AutoAttack, achieving detection rates consistently above 92% on CIFAR-10 and underscoring that intrinsic dimensionality is a powerful fingerprint for adversarial detection across diverse datasets and attack strategies.

## 1 Introduction

Deep Neural Networks (DNNs), despite their remarkable success, are notoriously vulnerable to adversarial attacks: small, carefully crafted perturbations to input data that can cause drastic misclassifications (Szegedy et al., 2013; Goodfellow et al., 2014). This vulnerability poses significant safety concerns for deploying DNNs in high-stakes domains such as medical diagnosis and autonomous driving, where robustness is non-negotiable (Eykholt et al., 2018). A primary defense strategy is to detect and discard adversarial inputs before they reach the model. However, reliably distinguishing adversarial examples from legitimate ones remains a significant open challenge.

Existing detection methods can be broadly categorized. On one hand, *distribution-based* methods analyze the statistical properties of data batches to identify outliers (Cui et al., 2023; Zhang et al., 2022). While often grounded in solid theory, they typically incur high computational costs and are ill-suited for real-time, single-sample detection. On the other hand, *perturbation-based* methods focus on intrinsic properties of individual samples, often defining a metric to identify abnormalities caused by adversarial noise (Feinman et al., 2017). These methods are computationally efficient but tend to be heuristic, relying on empirical validation and lacking a universally optimal metric.

A more promising direction lies in analyzing the geometry of the input loss landscape. It has been observed that natural and adversarial examples occupy geometrically distinct regions: natural inputs tend to reside in wide, flat valleys of the loss surface, whereas adversarial inputs are often found in narrow, steep regions (Zheng et al., 2023). This phenomenon arises because an attack perturbs a sample just enough to cross a decision boundary into a high-error, high-curvature area. While this geometric intuition is powerful, its practical application has been limited by the lack of a robust metric to quantify this "sharpness" effectively.

In this work, we propose that **Intrinsic Dimensionality (ID)** can serve as this missing quantitative measure. ID captures the minimal number of coordinates needed to describe local data on its underlying manifold, providing a precise geometric fingerprint of the loss landscape's curvature. We

hypothesize and empirically demonstrate that the perturbation introduced by an attack consistently alters the ID, making it a powerful criterion for detection.

Our main contributions are threefold:

1. We are the first to propose Intrinsic Dimensionality (ID) as a formal, quantitative metric to measure the sharpness of the input loss landscape for the purpose of adversarial detection.

2. We design a novel and efficient ID-based algorithm capable of detecting adversarial samples in both batch-wise and, critically, single-instance settings.

3. Through extensive experiments, we demonstrate that our method establishes a new state-of-the-art, significantly outperforming existing detectors against a wide array of powerful attacks on challenging benchmarks.

## 2 RELATED WORK

### 2.1 INTRINSIC DIMENSIONALITY IN ADVERSARIAL ROBUSTNESS

Intrinsic Dimensionality (ID) quantifies the minimum number of local coordinates required to describe data, effectively measuring the dimension of the manifold on which the data lies. High-ID regions are often diffuse and less stable, whereas low-ID regions are more compact and robust.

The connection between ID and adversarial robustness was first formally established by Amsaleg et al. (2017). Using a Maximum Likelihood Estimator (MLE) over nearest-neighbor distances, they proved that for $k$-NN classifiers, the expected perturbation magnitude needed to cause a misclassification is inversely proportional to the ID. This seminal result provides a formal link between geometric complexity and adversarial vulnerability: as ID increases, a sample's effective margin of safety shrinks.

Ansuini et al. (2019) extended this geometric perspective to deep neural networks. By applying estimators to the internal activations of major CNN architectures, they observed that ID typically peaks in mid-network layers before declining towards the classifier head. Crucially, they demonstrated that a lower final-layer ID correlates with higher generalization performance, positioning ID as a valuable diagnostic for representation quality.

Building on these foundations, Ma et al. (2018) introduced Local Intrinsic Dimensionality (LID), a point-wise estimator derived from extreme-value theory. They reported a consistent statistical gap, where adversarial examples exhibit a significantly higher LID than their benign counterparts in hidden layer representations. While they leveraged this insight to build a strong detector, their method requires access to intermediate activations, limiting its practicality for black-box models or at inference time. Collectively, these studies confirm that feature-space ID is a potent signal for distinguishing natural from adversarial data, but they leave open the question of whether an equally discriminative signal exists in the parameter-gradient space.

### 2.2 GRADIENT AND LOSS-GEOMETRY APPROACHES

A complementary line of research analyzes signals derived from the input-loss landscape and its gradients. For instance, Huang et al. (2021) proposed GradNorm, a detector that uses the $l_2$-norm of the parameter gradients, based on the observation that adversarial inputs induce larger gradient norms. While simple and effective, GradNorm condenses the rich gradient information into a single scalar, which can be sensitive to noise and may not capture the full geometric picture.

More recently, Zheng et al. (2023) adopted a direct landscape-centric view. They provided evidence that adversarial examples inhabit sharp, narrow minima, while natural inputs occupy broader, flatter basins—a contrast visualized in Figure 1. By designing a detector that explicitly estimates this local curvature, they confirmed that the landscape's geometry is a highly separable feature.

Our work unifies these two promising but separate lines of research. The studies above confirm that parameter gradients carry discriminative information and that loss landscape curvature is a key indicator of adversariality. We investigate whether the intrinsic dimensionality of the gradient space itself can serve as a more powerful and unified metric, simultaneously capturing the large-norm signals and the sharp-valley geometry to create a state-of-the-art detector.

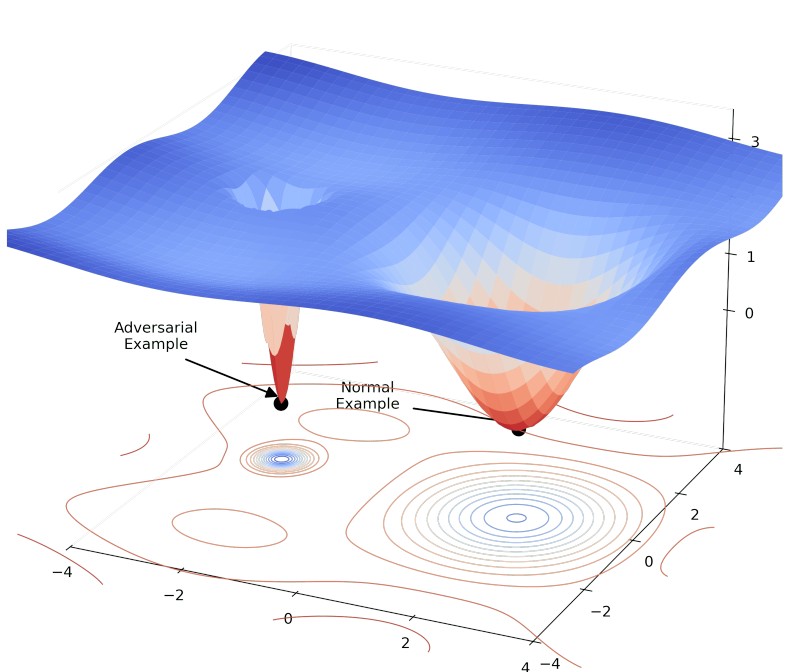

Figure 1: A visualization of the input loss landscape illustrating the geometric difference between normal and adversarial examples. Adversarial examples typically lie in sharp, narrow minima, whereas normal examples are found in wider, flatter basins. This visualization is adapted from the concept presented by Zheng et al. (2023).

## 3 BACKGROUND AND CORE HYPOTHESIS

While modern datasets reside in a high-dimensional ambient space (e.g., an image as a vector in $\mathbb{R}^n$), their core structure is often confined to a lower-dimensional manifold. The **intrinsic dimension (ID)** quantifies this effective dimensionality, representing the minimal number of parameters needed to describe the data's local structure without significant information loss. A classic illustration is a crumpled sheet of paper in 3D space: while any point on it requires three coordinates globally, the points themselves are constrained to a 2D surface—its intrinsic dimension.

Formally, the local ID at a point $x_i$ can be defined based on the rate at which the number of neighboring data points $N_i(r)$ scales within a small radius $r$ (Grassberger & Procaccia, 1983):

$$\text{ID}(x_i) = \lim_{r \to 0} \frac{\log N_i(r)}{\log(r)} \tag{1}$$

Since the true ID is unknown for real-world data, it must be estimated. In this work, we employ two well-established local estimators: the Maximum Likelihood Estimator (MLE) (Levina & Bickel, 2004) and the Two-Nearest-Neighbors (TwoNN) algorithm (Facco et al., 2017).

Prior research has leveraged ID to analyze the geometry of the **input space**. A key finding is that adversarial examples tend to exhibit a *higher* local ID than their natural counterparts (Ma et al., 2018). The intuition is that adversarial perturbations push samples into more complex, "brittle" regions of the input manifold, thus increasing the local geometric complexity.

In this work, we pivot from the geometry of inputs to the geometry of **parameter-gradients**. Our central thesis is that the sharp, narrow loss valleys associated with adversarial examples impose a strong structural constraint on the model's gradients. This constraint forces the gradient vectors,

$\nabla_\theta L(\theta; x, y)$, generated from adversarial inputs to occupy a highly correlated and therefore *lower-dimensional* subspace compared to gradients from natural inputs. This leads to a discernible disparity that is opposite to what is observed in the input space. We formally hypothesize that for a set of gradients from natural inputs, $G_{\text{natural}}$, and from adversarial inputs, $G_{\text{adversarial}}$, their respective intrinsic dimensions will consistently satisfy:

$$\text{ID}(G_{\text{natural}}) > \text{ID}(G_{\text{adversarial}}) \tag{2}$$

This shift in perspective forms the foundation of our detection method, providing a clear, quantifiable, and architecture-agnostic criterion for distinguishing adversarial examples.

## 4 PROPOSED METHOD

### 4.1 CONCEPTUAL FRAMEWORK

Our methodology is designed to differentiate adversarial inputs from benign ones by analyzing the geometric structure of the model's parameter-gradient space. While traditional methods focus on input-space perturbations or confidence scores, we posit that the **intrinsic dimension (ID)** of the gradient embeddings provides a more robust and model-agnostic signature of adversariality. Our core inquiry shifts from asking, *"What is the magnitude of the loss change?"* to *"How constrained is the subspace of parameter responses?"*

Building on the observation that adversarial examples occupy sharp, high-curvature minima in the input-loss landscape (Zheng et al., 2023), we hypothesize this localized sharpness forces the corresponding parameter gradients into a constrained, lower-dimensional subspace. Intuitively, for a model to react to a tiny input change with a large loss shift, its parameters must update along highly specific, coordinated axes. In contrast, natural samples from flatter regions of the loss surface permit more diffuse, less constrained gradient responses, which occupy a higher-dimensional space. This hypothesized disparity, $\text{ID}(G_{\text{natural}}) > \text{ID}(G_{\text{adversarial}})$, forms a clear, quantifiable criterion for detection that requires no architectural modifications or additional training.

### 4.2 GRADIENT EMBEDDING COMPUTATION

**Model Setup.** Our framework is model-agnostic. For empirical validation, we employ standard architectures such as ResNet-50 and ResNet-18 (He et al., 2016), pretrained on ImageNet. We replace the final fully-connected layer to match the target task and fine-tune all layers on the clean training set until convergence. This setup ensures our findings are generalizable across representative modern vision models.

**Loss and Gradients.** Let $f_\theta(x)$ be the network's softmax output for input $x$. We use the standard cross-entropy loss for the true label $y$: $L(\theta; x, y) = -\log[f_\theta(x)_y]$. For each sample $(x, y)$, we compute the parameter gradient $g(x, y) = \nabla_\theta L(\theta; x, y)$, which yields a vector in $\mathbb{R}^P$, where $P$ is the number of trainable parameters. To improve computational efficiency, we typically compute gradients only with respect to the parameters of the final layer(s), which we found preserves the discriminative signal while significantly reducing memory and runtime costs.

**Data Pipelines.** Our clean (*natural*) dataset consists of held-out validation images. For each clean image $x_i$ with label $y_i$, an adversarial counterpart $x_i + \delta_i$ is generated. This process yields two corresponding sets of gradient embeddings:

$$G_{\text{natural}} = \{\, g(x_i, y_i)\,\}_{i=1}^N, \quad \text{and} \quad G_{\text{adversarial}} = \{\, g(x_i + \delta_i, y_i)\,\}_{i=1}^N$$

**Batch vs. Single-Sample Mode.** We evaluate our detector in two distinct scenarios. In *batch mode*, we compute ID over a group of gradient embeddings to make a collective decision, suitable for identifying a malicious data source. In *single-sample mode*, each input is processed independently to provide a per-instance decision, offering maximum sensitivity to localized geometric anomalies.

### 4.3 INTRINSIC DIMENSION ESTIMATION

**Estimators and Parameters.** We estimate the ID of a set of gradient embeddings using two well-established algorithms. The first is the Two-Nearest Neighbors (TwoNN) estimator (Facco et al.,

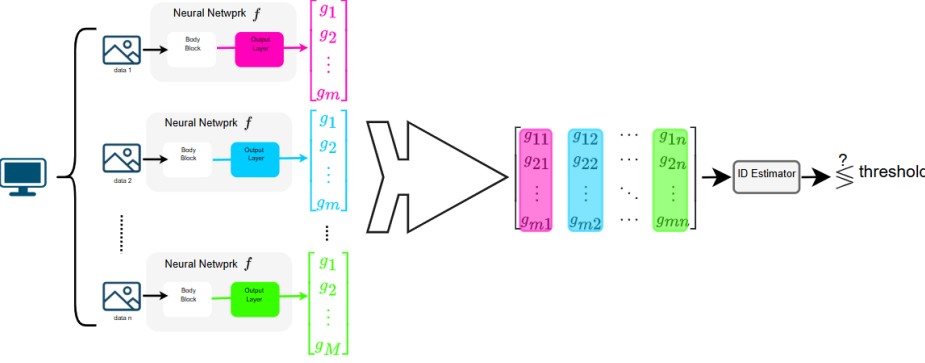

Figure 2: An overview of our batch-wise adversarial detection pipeline. A batch of $n$ input samples is processed by a neural network $f$. For each sample, the gradient vector $g$ of the loss with respect to the output layer's parameters is computed. These $n$ gradient vectors are aggregated into a single set, whose intrinsic dimension (ID) is then estimated. Finally, the resulting ID value is compared against a pre-determined threshold to classify the entire batch as either natural or adversarial.

2017), which computes dimension from the ratio of distances to the first two nearest neighbors. The second is the Maximum Likelihood Estimator (MLE) (Levina & Bickel, 2004), which generalizes this over the $k$ nearest neighbors. Following standard practice, we set $k = 10$ for MLE and average the estimate over bootstrap samples to ensure stability.

**Computational Procedure.** The full estimation process for determining the ID of a set of gradients (e.g., $G_{\text{natural}}$) is as follows:

1. For each sample in the set, compute its gradient embedding vector, restricted to the final layer(s).

2. For each embedding, find its $k$ nearest neighbors within the set using Euclidean distance.

3. Apply the chosen estimator's formula (TwoNN or MLE) to these neighbor distances to obtain a local ID estimate for that point.

4. Aggregate the local estimates (typically by averaging) to produce the final ID for the entire set.

**Efficiency Considerations.** To accelerate this process, especially for large datasets or models, several optimizations can be applied. These include estimating ID on a random subset of gradient vectors, employing approximate nearest-neighbor search libraries like FAISS (Johnson et al., 2019), and distributing distance computations across multiple GPU cores. These techniques can substantially reduce runtime with minimal impact on estimation accuracy.

## 5 EXPERIMENTS

To validate our hypothesis, we conduct a series of experiments designed to evaluate our ID-based detection method across diverse datasets, attack methodologies, and operational scenarios.

### 5.1 BATCH-WISE GRADIENT ANALYSIS

**Setup and Datasets.** We first consider a setting inspired by Federated Learning (McMahan et al., 2017), where a central server must validate gradient updates from multiple clients. Malicious clients may submit gradients computed from adversarial examples. Our goal is to identify these clients. Figure 2 provides a high-level overview of our detection pipeline for this scenario. As detailed in Algorithm 1, the server computes the ID of a trusted reference set of natural gradients ($ID_{\text{natural}}$) and compares it to the ID of the incoming batch from each client ($ID_k$). A client is flagged as adversarial if the deviation $|ID_k - ID_{\text{natural}}|$ exceeds a threshold $\tau$. We simulate this scenario with $K = 5$ clients on **SVHN** (Netzer et al., 2011), **MNIST** (LeCun et al., 1998), and **CIFAR-10**

---

**Algorithm 1** Adversarial Client Detection via Intrinsic Dimensionality

---

**Require:** Global model $f_\theta$, loss $L$, client datasets $\{D_k\}_{k=1}^K$, estimator estimate_id, threshold $\tau$,
    reference dataset $D_{\text{natural}}$
**Ensure:** Client labels $\{\text{ClientType}_k\}$
  1: $G_{\text{natural}} \leftarrow \{\nabla_\theta L(\theta; x_i, y_i)\}_{(x_i, y_i) \in D_{\text{natural}}}$
  2: $ID_{\text{natural}} \leftarrow \text{estimate\_id}(G_{\text{natural}})$
  3: **for** $k = 1$ to $K$ **do**
  4:      $G_k \leftarrow \{\nabla_\theta L(\theta; x_{k,i}, y_{k,i})\}_{(x_{k,i}, y_{k,i}) \in D_k}$
  5:      $ID_k \leftarrow \text{estimate\_id}(G_k)$
  6:      **if** $|ID_k - ID_{\text{natural}}| \leq \tau$ **then**
  7:          $\text{ClientType}_k \leftarrow$ natural
  8:      **else**
  9:          $\text{ClientType}_k \leftarrow$ adversarial
10:      **end if**
11: **end for**
12: **return** $\{\text{ClientType}_k\}$

---

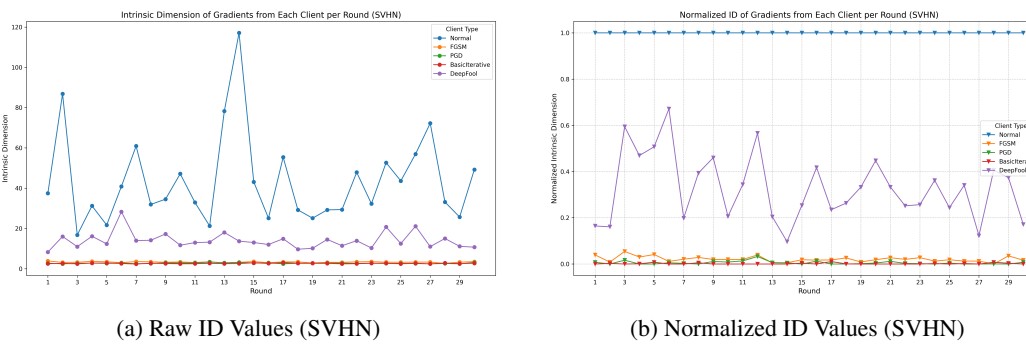

(a) Raw ID Values (SVHN)          (b) Normalized ID Values (SVHN)

Figure 3: Batch-wise detection results on SVHN. The benign client (Normal) consistently exhibits a different intrinsic dimension from the four malicious clients, enabling robust detection.

(Krizhevsky & Hinton, 2009) using a ResNet-50 model. One client remains benign, while the other four use FGSM (Goodfellow et al., 2014), PGD (Madry et al., 2017), BIM (Kurakin et al., 2016), and DeepFool (Moosavi-Dezfooli et al., 2016) attacks, respectively.

**Results.** We compare our ID-based detector against average gradient norm and confidence score baselines. As shown in Figure 3 for the SVHN dataset, our method achieves a clear and consistent separation between the ID of the benign client's gradients and those of the four malicious clients. This distinct geometric gap enables highly accurate detection (over 95% accuracy in simulations), whereas baselines often failed to distinguish clients due to significant overlap in their metrics. This trend holds across all datasets; full results are provided in the Appendix.

### 5.2 INDIVIDUAL SAMPLE ANALYSIS

**Setup and Datasets.** For safety-critical applications, real-time detection of individual adversarial samples is paramount. Our workflow (Figure 4) maintains a reference manifold of natural gradient embeddings, $G_{\text{norm}}$, and classifies an incoming sample $x^*$ based on how its gradient $g^*$ perturbs this manifold's geometry. As detailed in Algorithm 2, a sample is flagged if the ID of the augmented set, $\text{ID}(G_{\text{norm}} \cup \{g^*\})$, falls outside a percentile-based confidence interval derived from the natural distribution. We evaluate this on **SVHN** using a ResNet-18 against PGD (Madry et al., 2017) and AutoAttack (Croce & Hein, 2020).

**Results.** On SVHN, our method achieves a strong 85.4% overall detection accuracy against a mix of PGD and AutoAttack samples. This effectiveness stems from the clear geometric separability induced by adversarial gradients, as visualized in Figures 5 and 6. The ID distributions confirm

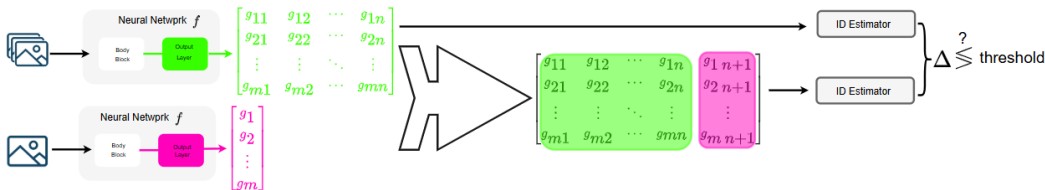

Figure 4: Workflow for per-sample adversarial detection. The method establishes a baseline intrinsic dimension ($ID_{\mathrm{natural}}$) from a reference set of $n$ natural gradient embeddings. For an incoming test sample, its gradient embedding is appended to the reference set. The sample is classified as adversarial if the change in ID of this augmented set ($\Delta = |ID_{\mathrm{aug}} - ID_{\mathrm{natural}}|$) surpasses a threshold.

---

**Algorithm 2** Per-Sample Adversarial Detection via ID Perturbation

---

**Require:** Model $f_\theta$, loss $L$, reference set $G_{\mathrm{norm}}$, test sample gradient $g^*$, estimator $\mathrm{estimate\_id}$
**Require: Parameters:** Pre-computed percentiles $P_{\mathrm{low}}, P_{\mathrm{high}}$ from the natural distribution of ID values
**Ensure:** Label $\in$ {natural, adversarial}
1: $ID_{\mathrm{norm}} \leftarrow \mathrm{estimate\_id}(G_{\mathrm{norm}})$
2: $G_{\mathrm{aug}} \leftarrow G_{\mathrm{norm}} \cup \{g^*\}$
3: $ID_{\mathrm{aug}} \leftarrow \mathrm{estimate\_id}(G_{\mathrm{aug}})$
4: **if** $ID_{\mathrm{aug}} \in [P_{\mathrm{low}}, P_{\mathrm{high}}]$ **then**
5:     **return** natural
6: **else**
7:     **return** adversarial
8: **end if**

---

that percentile-based thresholds effectively demarcate benign gradients from those generated by powerful attacks.

## 5.3 SOTA Comparison on CIFAR-10 & MS COCO

**Setup and Datasets.** We conduct a large-scale evaluation of our per-sample detector on **CIFAR-10** (Krizhevsky & Hinton, 2009) and a 4,952-image subset of **MS COCO 2017** (Lin et al., 2014). We compare against a suite of strong attacks detailed in Table 1. For CIFAR-10, we fine-tune a ResNet-18, while for MS COCO, we train a linear head on a frozen ResNet-18. We use the MLE estimator for CIFAR-10 (on 10D PCA-reduced gradients) and the TwoNN estimator for MS COCO. Thresholds are calibrated on 1,000 held-out clean images.

**Results.** Tables 2 and 3 present our main results, comparing our method against nine state-of-the-art detectors, with performance measured by the adversarial detection rate ($\mathrm{DR}_a = \frac{\mathrm{TP}}{\mathrm{TP+FN}}$). On CIFAR-10, our method achieves near-perfect detection against several attacks and consistently exceeds 92% across the board. On the more challenging, high-resolution MS COCO dataset, our detector maintains strong performance, with detection rates ranging from 85.4% to 95.3%.

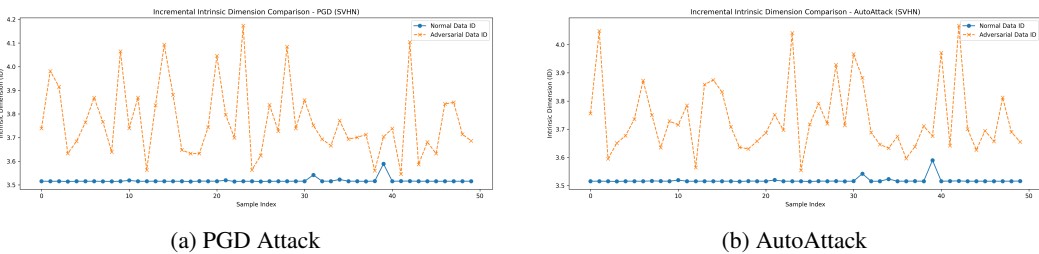

(a) PGD Attack          (b) AutoAttack

Figure 5: Per-sample ID analysis on SVHN. The ID of the augmented manifold deviates significantly when an adversarial sample's gradient is introduced.

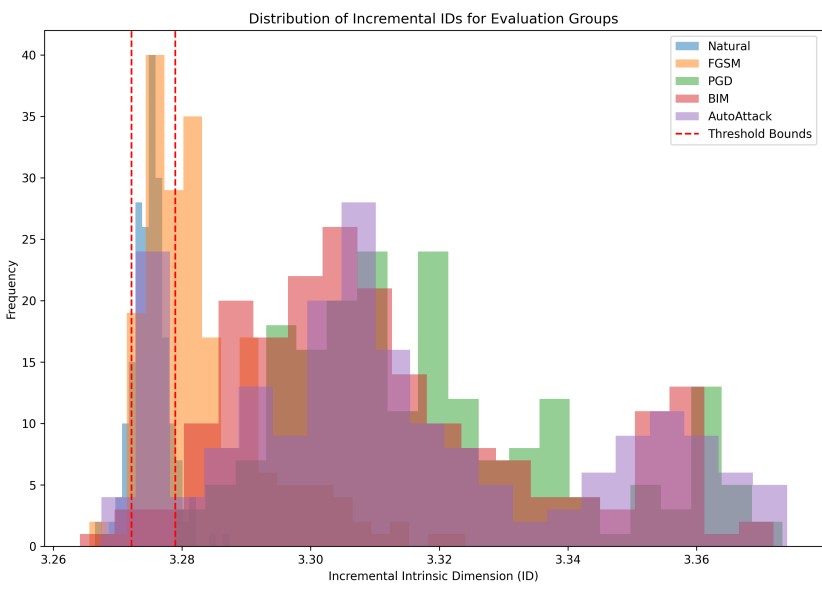

Figure 6: Distribution of the augmented set's ID on SVHN. Adversarial attacks consistently shift the ID relative to natural data, enabling effective separation via percentile thresholds (dashed lines).

Table 1: Adversarial Attack Parameters.

| Attack | Parameters |
|---|---|
| FGSM | $\epsilon = 0.008$ |
| PGD | $\epsilon = 0.01$, $\alpha = 0.02$, steps=40 |
| BIM | $\epsilon = 0.03$, $\alpha = 0.01$, steps=10 |
| DeepFool | steps=20 |
| CW ($L_2$) | $C = 2$, $\kappa = 2$, steps=500, lr=0.01 |

**Discussion.** Our ID-based detector consistently outperforms existing methods by a significant margin, particularly on CIFAR-10. The results demonstrate that the intrinsic dimensionality of the gradient space is a highly reliable and generalizable signal for adversarial detection. The method's robustness on both low-resolution (CIFAR-10) and high-resolution (MS COCO) data underscores its effectiveness as a practical defense mechanism.

Table 2: Adversarial Detection Rate (%) on CIFAR-10.

| Method | FGSM | PGD | BIM | DF | CW |
|---|---|---|---|---|---|
| SAC (Liu et al., 2022) | 60.1 | 59.7 | 56.8 | 21.6 | 17.7 |
| Sim-DNN (Soares et al., 2022) | 70.5 | 60.0 | 49.4 | 26.7 | 22.9 |
| DTBA (Qi et al., 2022) | 78.3 | 75.6 | 71.7 | 36.2 | 32.3 |
| MH-UI (Yang et al., 2023) | 79.2 | 76.5 | 74.6 | 49.1 | 52.5 |
| AAE (Ji et al., 2022) | 80.5 | 76.9 | 75.4 | 63.7 | 60.2 |
| HSJ (Hussain & Hong, 2023) | 77.5 | 75.2 | 75.6 | 60.1 | 59.4 |
| HM (Picot et al., 2023) | 86.9 | 84.5 | 84.0 | 80.6 | 77.9 |
| FCB (Iglesias et al., 2019) | 49.8 | 47.1 | 43.6 | 15.1 | 11.4 |
| MF (Jara et al., 2022) | 51.4 | 48.0 | 46.1 | 16.4 | 11.9 |
| MADM (Ranjbar & Effati, 2022) | 62.4 | 54.2 | 51.5 | 19.0 | 14.3 |
| **Ours** | **96.4** | **100.0** | **98.4** | **92.7** | **100.0** |

Table 3: Adversarial Detection Rate (%) on MS COCO.

| Method | FGSM | PGD | BIM | DF | CW |
|---|---|---|---|---|---|
| SAC (Liu et al., 2022) | 58.7 | 56.3 | 37.8 | 21.1 | 16.8 |
| Sim-DNN (Soares et al., 2022) | 63.5 | 74.1 | 37.8 | 24.2 | 22.8 |
| DTBA (Qi et al., 2022) | 74.6 | 79.8 | 37.8 | 34.0 | 31.6 |
| MH-UI (Yang et al., 2023) | 76.0 | 80.3 | 37.5 | 47.5 | 50.1 |
| AAE (Ji et al., 2022) | 77.3 | 82.0 | 69.1 | 56.5 | 55.1 |
| HSJ (Hussain & Hong, 2023) | 76.6 | 73.9 | 68.3 | 58.8 | 57.5 |
| HM (Picot et al., 2023) | 85.6 | 89.6 | 84.9 | 78.3 | 75.8 |
| FCB (Iglesias et al., 2019) | 46.7 | 51.1 | 14.4 | 14.3 | 10.5 |
| MF (Jara et al., 2022) | 48.8 | 56.4 | 17.7 | 15.9 | 11.5 |
| MADM (Ranjbar & Effati, 2022) | 61.2 | 64.8 | 22.8 | 18.5 | 14.0 |
| **Ours** | **93.9** | **95.3** | **86.2** | **85.4** | **87.6** |

## 6 CONCLUSION

In this work, we demonstrated that the intrinsic dimension (ID) of the parameter-gradient space serves as a powerful and robust signal for adversarial detection. Our central hypothesis, confirmed through extensive experiments, is that gradients generated from adversarial examples inhabit a manifold of significantly lower intrinsic dimension than those from natural examples. By leveraging this geometric disparity, our method establishes a new state-of-the-art on challenging benchmarks, including CIFAR-10 and MS COCO. Operating in both batch-wise and single-sample modes, it achieves detection rates consistently above 92% on CIFAR-10 against a wide range of attacks.

While our method shows strong empirical success, we identify two primary areas for future investigation. The first is the computational overhead of per-sample ID estimation. Future work should explore more efficient or approximate ID estimators to facilitate real-time, low-latency deployment. The second is robustness against adaptive attacks. A crucial next step is to evaluate our detector against adversaries specifically designed to preserve the gradient manifold's geometry, which will be key to developing a truly resilient defense.

## REPRODUCIBILITY STATEMENT

To ensure the reproducibility of our results, we have made our source code, including scripts for data processing, model training, and evaluation, available as part of the supplementary material. The code is implemented in Python using the PyTorch framework. The supplementary material also includes details on the specific package versions used in our environment. All datasets used in this paper (MNIST, SVHN, CIFAR-10, and MS COCO) are publicly available. We believe this provides all the necessary components for our results to be fully reproduced.

## ETHICS STATEMENT

This research focuses on defending against adversarial attacks, a critical aspect of ensuring the safety and reliability of machine learning systems. While our work is defensive in nature, we acknowledge that any research into adversarial phenomena could potentially be misused by malicious actors. We have chosen not to release code for generating new or more powerful attacks. Our primary contribution is a detection method, which we believe contributes positively to the development of more robust and trustworthy AI. We have conducted all experiments on publicly available datasets and foresee no direct negative societal consequences from this work.

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
