# OpenReview forum: "Gradient Manifold Geometry as a Signature for Adversarial Detection"
_ICLR.cc/2026/Conference — ICLR 2026 Conference Withdrawn Submission_

### Official Review · Reviewer_dhVy · 2025-10-30

**Soundness:** 2
**Presentation:** 3
**Contribution:** 3
**Rating:** 2
**Confidence:** 3

**Summary:**

The paper proposes a new adversarial detection method based on the intrinsic dimensionality (ID) of the gradient manifold. The key hypothesis is that adversarial examples produce gradients that lie in a more constrained, lower-dimensional subspace compared to natural examples, due to sharper and narrower regions of the loss landscape.

**Strengths:**

1) Using the intrinsic dimensionality (ID) of the gradient manifold as a geometric signature for adversarial detection is an interesting and novel idea that introduces a fresh geometric perspective to the problem.

2) The paper is clearly written and easy to follow.

3) The authors provide the implementation code in the supplementary material.

**Weaknesses:**

1) The paper references an Appendix section for additional results (e.g., at line 310: “full results are provided in the Appendix”), but no appendix was included in the submission. The results are provided only for the SVHN dataset.

2) The proposed detector assumes access to extensive information, including model parameters, true labels, and even a reference set of natural gradients, which is unrealistic. Although it may not require knowing the exact architecture (e.g., ResNet vs. DenseNet), it still assumes access to the model’s internal parameters to compute gradients.

3) The experimental evaluation is weak. There is no discussion of the threshold parameter selection or sensitivity, nor of the attacking strategy. The study includes only five standard attacks without varying their magnitude or norm and provides no evidence of their actual effectiveness. The evaluation is limited to ResNet architectures, with no results on CIFAR-100 or TinyImageNet. Finally, the paper lacks any discussion or experiments on adaptive attacks, which are essential to assess detector capacity.

**Questions:**

1) The method assumes a well-behaved model, but how does the intrinsic dimensionality behave when the classifier is miscalibrated or overfitted? For instance, if the network exhibits overconfidence or poor generalization on the clean set, do the gradient manifolds of natural and adversarial samples still show a consistent ID gap?

2) The paper estimates ID in the gradient space using Euclidean distances to compute nearest neighbors. However, given that gradient embeddings lie in very high-dimensional spaces where Euclidean distances tend to concentrate and may not reflect meaningful geometric relationships, could the authors clarify or justify this choice?

3) In Section 5.2, the per-sample detector computes $g(x,y)$, but the true label $y$ is not available at test time. If $y$ is known, detecting adversarial samples becomes redundant. How is the detection score computed in this setting? If it is used, then it is a severe limitation of the method, and the results in Table 2 are not surprising.

4) The paper does not discuss how the decision threshold for ID deviation is selected or tuned. How sensitive is the detection performance to this parameter, and how is it calibrated across datasets or attack types? No results in terms of AUROCs are provided.

---

### Official Review · Reviewer_u14d · 2025-10-31

**Soundness:** 2
**Presentation:** 2
**Contribution:** 2
**Rating:** 4
**Confidence:** 3

**Summary:**

The paper proposes using the intrinsic dimensionality (ID) of the parameter-gradient space as a geometric signature for detecting adversarial examples. The authors hypothesize that adversarial inputs constrain the model’s gradients to a lower-dimensional subspace compared to natural samples. They estimate ID using standard algorithms (TwoNN, MLE) and show empirical results on both batch-level and single-sample detection tasks across datasets (MNIST, SVHN, CIFAR-10, MS COCO). The method achieves strong detection accuracy, outperforming prior baselines.

**Strengths:**

The paper presents a novel and intriguing perspective on adversarial detection by shifting the analysis from the commonly studied input space to the parameter space. While most prior work focuses on characterizing the geometry of input or feature representations, exploring the gradient and weight landscape as a discriminative signal is an original and thought-provoking direction. This conceptual shift could open new avenues for understanding how model parameters encode robustness.

**Weaknesses:**

1. Conceptual inconsistencies:
The loss landscape is defined in weight space, not input space, so the geometric intuition between them is blurred. Loss values and gradients are computed using true labels, it is unclear how to translate it to adversarial samples. In Figure 1 the natural and adversarial image are not optimizing the same loss.

2. Dimensionality discussion lacks grounding:
Early sections mix data-space and weight-space dimensionality without clear theoretical justification or related work connecting the two.
“Gradient space” is not formally distinct from just parameter space.

3. Federated learning scenario feels ad hoc: not well-motivated, no relevant related work or comparisons to federated-learning defenses. In this setting the true label is known.

4. Assumption on true label knowledge: The method computes gradients using the true labels, even for adversarial examples. This assumption weakens the practical relevance of the approach, as in real-world scenarios the true label of an input is unknown at inference time. If such information were available, a simpler solution as direct inference would already suffice for detection.

5. Experimental limitations:
Tested mainly on small models and low-dimensional datasets—unclear scalability or efficiency on larger systems. Not evaluated under adaptive white-box attacks, which could easily exploit or neutralize the geometric signal. Uses only final-layer gradients without ablation to justify why this subset captures the signal.

**Questions:**

1. How do the authors interpret the loss landscape in the input space, given that loss is defined with respect to a label?
2. Since adversarial examples are typically misclassified, what does it mean to compute the loss (and its gradient) using the true label, which is unknown at inference time? Would the method still work if gradients were computed using the predicted label instead?
3. What is the motivation for using only the last-layer gradients rather than the full model? Was an ablation study performed to justify this choice?
4. Was the proposed method evaluated under adaptive white-box attacks specifically designed to fool the detector? If not, how do the authors expect the method to perform in such settings?

---

### Official Review · Reviewer_53bG · 2025-11-01

**Soundness:** 1
**Presentation:** 2
**Contribution:** 1
**Rating:** 0
**Confidence:** 5

**Summary:**

This paper proposes a novel method for detecting adversarial examples by utilizing the Intrinsic Dimensionality (ID) of gradient parameters as a signature. Experimental evaluations conducted under two distinct scenarios demonstrate the partial effectiveness of the proposed approach.

**Strengths:**

This paper proposes a novel perspective that leverages the geometric information of model parameter gradients—unlike most existing methods, which focus solely on the geometry of input examples.

**Weaknesses:**

**1**. This work lacks rigor, as many viewpoints and statements are insufficiently supported by evidence or explanation. For example:

+ **Line 71**: The claim that "High-ID regions are often diffuse and less stable, whereas low-ID regions are more compact and robust" requires substantiation. The authors should provide supporting evidence and clarify their reasoning.

+ **Line 98**: The assertion that "condensing the rich gradient information into a single scalar would be sensitive to noise and may not capture the full geometric picture" is insufficiently justified. A more detailed explanation is needed.

+ **Line 161**: The central thesis—"the sharp, narrow loss valleys associated with adversarial examples impose a strong structural constraint on the model’s gradients"—is critical to the proposed methods’ effectiveness, yet its validity remains unproven. The authors must demonstrate the reasonableness of this assumption with compelling evidence.

+ **Line 184**: The statement—"For a model to react to a tiny input change with a large loss shift, its parameters must update along highly specific, coordinated axes"—is not intuitively obvious. Further elaboration is warranted.

+ **Line 186**: Similarly, the claim that "natural samples from flatter regions of the loss surface permit more diffuse, less constrained gradient responses, which occupy a higher-dimensional space" requires additional clarification and justification.


**2**. Using intrinsic dimensionality (ID) to distinguish between normal and adversarial samples is a well-established approach. While the authors’ method—measuring intrinsic dimension with respect to the model’s loss gradients in parameter space—introduces novelty, its necessity and advantages over conventional approaches (e.g., computing ID in input sample space) remain unclear.

**3**. This work examines a scenario for detecting malicious clients in federated learning, where attackers revert to uploading gradients derived from adversarial examples. This assumption raises several concerns:

+ **3.1 Validity of the assumption**: Is this scenario meaningful and realistic, or is it purely hypothetical?

+ **3.2 Practicality of adversarial example generation**: Even if we accept this scenario as meaningful, how exactly would a malicious client generate adversarial examples? If we assume these examples are designed to target the federated learning model's final output, this premise appears fundamentally flawed. Conversely, if these aren't adversarial examples aimed at the federated learning model, how can we ensure they meet the proposed method's assumptions? Without knowledge of the model's loss function, this guarantee becomes particularly problematic.

+ **3.3 Fundamental inconsistency**: The primary issue in this scenario is that the gradient uploaded by the malicious client deviates entirely from the gradient of the model loss with respect to the parameters assumed by the proposed method. Specifically, one gradient corresponds to the inference stage (when the model is already trained), whereas federated learning involves gradients during the training stage. Consequently, experiments conducted under these conditions do not adequately validate the effectiveness of the proposed method. Moreover, from a fundamental standpoint, the proposed method is ill-suited for detecting malicious clients in the first place.

**4**. The experimental design of this study suffers from several limitations.

+ 4.1: The chosen metrics fail to comprehensively evaluate the method's performance. While the authors rely solely on the Attack Detection Rate (ADR, i.e., recall), standard practice in adversarial example detection necessitates broader binary classification metrics such as the AUROC score and F1 score.

+ 4.2: The choice of comparison algorithms is inappropriate as it overlooks several established benchmarks in the field of adversarial example detection, including LID [1], MD [2], SID [3], and GradNorm [4]. While the authors acknowledge the relevance of these methods in the Related Work section, they fail to include them in their comparative analysis. Moreover, the selected comparison algorithms do not adequately represent an appropriate benchmark for this work. **For instance, DTBA (mentioned in Table 2) is actually an adversarial attack method—not a detection technique—raising questions about why it was chosen as a baseline comparison.** Due to these issues, this paper has not sufficiently demonstrated the effectiveness of the proposed method through a proper comparative analysis.

+ 4.3: The paper lacks discussion regarding several key hyperparameters in the proposed method. For instance, the selection of $P_{\text{low}}$ and $P_{\text{high}}$.

+ 4.4: Current adversarial detection methods often fail to account for naturally noisy samples, which are highly susceptible to being misclassified as adversarial. This work does not address this limitation.

[1] Xingjun Ma, Bo Li, Yisen Wang, Sarah M Erfani, Sudanthi Wijewickrema, Grant Schoenebeck, Dawn Song, Michael E Houle, and James Bailey. Characterizing adversarial subspaces using local intrinsic dimensionality. In International Conference on Learning Representations, 2018.

[2] Lee, K.; Lee, K.; Lee, H.; and Shin, J. 2018. A Simple Unified Framework for Detecting Out-Of-Distribution Samples
and Adversarial Attacks. In Advances in Neural Information Processing Systems, 7167–7177.

[3] Jinyu Tian, Jiantao Zhou, Yuanman Li, Jia Duan, Detecting adversarial examples from sensitivity inconsistency of spatial-transform domain, In Proceedings of the AAAI Conference on Artificial Intelligence, 2021.

[4] Xue-Yuan Huang, Yu-Kun Zhang, Yong-Fei Dou, and Bing Liu. GradNorm: A gradient-based regularizer for out-of-distribution detection. In Proceedings of the AAAI Conference on Artificial Intelligence, volume 35, pp. 7434–7442, 2021.

**Questions:**

See the weakness.

---

### Official Review · Reviewer_BJZP · 2025-11-03

**Soundness:** 3
**Presentation:** 3
**Contribution:** 4
**Rating:** 6
**Confidence:** 3

**Summary:**

This paper proposes a new adversarial attack detection method based on the geometry of the model’s parameter gradient. In particular, this paper utilizes the concept of the intrinsic dimension (ID) to analyze the local geometry of the loss gradient manifold. The authors hypothesize that the intrinsic dimension of local gradients can serve as an indicator of adversariality. The proposed method first estimates the intrinsic dimension for a given input, then it measures the gap between the estimated intrinsic dimension and a precomputed value derived from benign samples. The paper applies the defense to two different practical scenarios: A batch-wise analysis representing a federated learning setting and a common scenario that analyzes individual samples. Through experiments, the authors validate their hypothesis on the intrinsic dimensions of adversarial examples and demonstrate the effectiveness of the detection method.

**Strengths:**

1. To the best of my knowledge, the proposed method is novel, and this is the first paper that exploits the geometry of the **gradient manifold**.
2. The experiments involve various setups, differing in the practical context, the number of datasets, and the number of attacks.
3. The proposed detector demonstrates impressive performance.

**Weaknesses:**

The model architectures in the experiments are limited to the ResNet variants.

**Questions:**

1. Consider adding more experiments with model architectures other than ResNet variants, e.g., Vision Transformer.
2. How expensive is the ID computation (including the nearest neighbor finding), especially in the batch-wide setting?
3. DeepFool has induced relatively large ID values. Can the authors explain why? Does it also imply that some modification of the DeepFool may generate adversarial examples that can circumvent the proposed detector?

---

### Note · Authors · 2025-11-27

**Comment:**

We respectfully withdraw this submission.

**Withdrawal Confirmation:**

I have read and agree with the venue's withdrawal policy on behalf of myself and my co-authors.